# Mortality rates in a cohort of infants attending immunization clinics in Uganda (2017–2019)

Samuel Sendagala[1]*, Rose Bosa Nakityo[1], Fredrick Makumbi[2], Tom Lutalo[2], Linda Nabitaka[3], Fred Nalugoda[2], Ivan Lukabwe[3], Jingo Kasule[2], Emily Namara-Lugolobi[1], Margaret Achom Okwero[1], Hilda T. Asiimwe[4], Phoebe Namukanja[1], Bernadette Ng'eno[5], Emilio Dirlikov[1], Augustina Delaney[5]

**1** US Centers for Disease Control and Prevention, Global Health Center, Division of Global HIV and TB, Kampala, Uganda, **2** Data and Statistics Department, Rakai Health Sciences Program, Kalisizo, Uganda, **3** STI/AIDS Control Program, Department of National Disease Control, Ministry of Health, Kampala, Uganda, **4** United States Agency for International Development, Kampala, Uganda, **5** US Centers for Disease Control and Prevention, Global Health Center, Division of Global HIV and TB, Atlanta, Georgia, United States of America

* hlw6@cdc.gov

## Abstract

### Background

Uganda reported a significant reduction in the mortality rate of children under 5 years of age, from 146/1,000 live births in 2000–42/1,000 live births in 2021. With the roll-out of Option B+, the vertical transmission rate of HIV decreased from 13.0% (2012) to 6.0% (2019). However, its impact on the mortality rate among children is not well documented. We determined the mortality rate and associated risk factors among infants exposed and not exposed to HIV attending immunization clinics in Uganda.

### Methods

We conducted an observational prospective cohort study of mother–infant pairs (MIPs) with infants exposed or unexposed to HIV. We enrolled infants aged 4–12 weeks. The inclusion criteria were biological mothers attending health facilities that provide routine immunization for children and/or postnatal care visits who were able to provide signed written informed consent; mothers or infants who were not severely ill; and those who consented to have their infants tested for HIV antibodies at baseline and follow-up visits every 3 months until the children were aged 18 months. Child-HIV infection and death were censored events. Children lost to follow-up or withdrawn from the study were censored from analyses at the last documented study visit. The outcome of interest was child mortality, and the independent variables were mother's age; infant HIV exposure status; infant sex; family socioeconomic status; marital status; education level; malaria during pregnancy; birth attendee; mother's ART initiation; mode of transport to health facilities; breastfeeding pattern; 4 or more

**Data availability statement:** All relevant data are within the paper and its Supporting Information files.

**Funding:** This research has been supported by the President's Emergency Plan for AIDS Relief (PEPFAR) through the Centers for Disease Control and Prevention (CDC) under the terms of cooperative agreement number 5U2G GH000817-02.

**Competing interests:** The authors have declared that no competing interests exist.

ANC visits; and mother's baseline viral load nonsuppression and place of delivery. We used Kaplan–Meier survival curves to estimate cumulative mortality probability and the Wilcoxon log-rank test to compare differences in cumulative survival functions. We used multivariate Weibull proportional hazards and Weibull accelerated failure time (AFT) regression models with 95% confidence intervals (CIs) to identify factors associated with child death.

## Results

Among the 16,718 MIPs identified, 11,519 (68.9%) mothers consented to study follow-up. At the 18-month follow-up, 0.7% (79/11,519) of the infants had died, 40.5% (32/79) of whom were exposed to HIV. The overall child mortality rate per 1,000 person-years was 5.0 (95% CI: 4.0--6.2) and was significantly greater among the infants exposed to HIV (14.2; 95% CI: 10.0--20.0) than among the infants not exposed to HIV (3.5; 95% CI: 2.6--4.6). In the adjusted model, the mortality risk factors were HIV exposure status (aHR 5.6 95% CI: 3.5--9.4), maternal age < 25 years (aHR 1.8; 95% CI: 1.1--2.9), living without a partner (aHR 1.8; 95% CI: 1.1--2.9), and delivery at home (aHR 2.2; 95% CI: 1.3--4.0).

## Conclusion

Single young mothers living with HIV delivering at home increased the risk of child mortality. Identifying mothers with risk factors early for support could reduce the risk of child mortality.

## Introduction

In 2021, 5.0 million children died before they reached the age of 5 years, and 56% (2.8 million) were in sub–Saharan Africa, despite the region accounting for 29% of the global live births [1]. There has been a 59% reduction in the child underfive mortality rate (U5MR) globally in previous decades, from 93 to 38 deaths per 1,000 live births between 1990 and 2021. However, the reduction in the U5MR in sub-Saharan Africa (SSA) was slower than that in other regions, with 74 deaths per 1,000 live births in 2021 [1]. The implementation of the United Nations' Sustainable Development Goals (SDGs) started in 2016 with a target U5MR of no more than 25 deaths per 1,000 live births in every country of the world by 2030 [2]. Uganda has yet to achieve this target, with a current U5MR of 42 deaths per 1,000 live births in 2021 [1].

Globally, communicable and infectious diseases such as pneumonia, diarrhea, HIV/AIDS, and malaria continue to be the leading causes of preventable underfive deaths [3]. Other factors, such as death of the mother, child vaccination status, breastfeeding status and low birth weight, have also been shown in several studies to be associated with morbidity and mortality among infants and are more pronounced among infants exposed than those not exposed to HIV [4–6]. Mosley et al.'s analytical approach to the study of the determinants of child survival in low- and

middle-income countries highlights the premise that all socioeconomic determinants, such as income/wealth, the political economy and the health system of child mortality, operate through a common set of biological mechanisms or proximate determinants, such as maternal factors, nutrition, and personal illness control, to affect mortality [7].

Exposure to HIV has been widely documented to be a risk factor for infant mortality in the ZVITAMBO trial, which reported a 2-year mortality rate of 9.2% in infants exposed but not infected compared with 2.9% in those not exposed to HIV [8,9]. However, maternal antiretroviral therapy (ART) use during pregnancy and breastfeeding has been shown to reduce mortality in children under 5 years of age to levels observed in children of mothers not infected with HIV in South Africa [10]. Young mother, illiteracy and income inequality have been documented to predict the infant mortality rate (IMR) in low- and middle-income countries [11]. In 2013, in accordance with the WHO guidelines, Uganda was among the first countries in sub-Saharan Africa to incorporate Option B+ (life-long universal maternal antiretroviral therapy (ART)) into its national prevention of vertical transmission of HIV strategy [12]; since then, the percentage of pregnant and breastfeeding women living with HIV receiving lifelong ART has increased from <1% to >95%, and transmission rates have decreased from 24% to 6% from 2010–2019 [12–17]. Uganda is a country with a mature generalized HIV epidemic, with an HIV prevalence of 5.8% among adults aged 15 years plus approximately 1.5 million people living with HIV [18]. Over time, the HIV epidemic has had a negative impact on overall child mortality rates in the country; thus, with increased access to ART for pregnant women living with HIV, the rate of HIV-related child mortality has been declining in Uganda. To determine child mortality rates and associated risk factors among infants exposed and not exposed to HIV, we analyzed data from the evaluation of the Impact of the national program for prevention of vertical transmission of HIV in Uganda, a nationally representative cohort of infants exposed and not exposed to HIV in Uganda up to 18 months postdelivery.

## Methods

We analyzed data collected as part of the Uganda prevention of vertical transmission of HIV impact evaluation study, a prospective observational cohort study of vertical transmission of HIV. The study included a nationally representative sample of mother–infant pairs (MIPs), including infants exposed and those not exposed to HIV, recruited from immunization clinics within health facilities from 20th September 2017–26th February 2018. The health facility sampling frame included all private and public facilities that offered infant first DPT vaccination in 2015, of which 206 health facilities were randomly selected without replacement. Infants aged 4–12 weeks and their biological mothers receiving routine health care at the selected facilities were screened. Trained research assistants obtained written consent and enrolled mother–infant pairs. Mother-infant pairs were excluded if the infant or mother was severely ill or if the mother did not consent to infant HIV testing. The infants were followed until incident HIV infection, death, or 18 months of age. MIPs had follow-up visits at 6, 9, 12, 15, and 18 months postpartum, with a ±6-week allowance for follow-up visits. All infants exposed to HIV were assessed for HIV infection at follow-up visits with infants aged <18 months via HIV DNA PCR via dried blood spots and those aged ≥18 months through onsite rapid antibody testing. For mothers living without HIV, at every visit, they were screened for new HIV infection via onsite rapid antibody testing. In addition, mothers living with HIV underwent maternal viral load testing at the 6- and 12-month follow-up visits. Data were obtained through maternal and infant questionnaires and abstraction of selected clinical variables from the child's Child Health Card (CHC).

Child characteristics collected at baseline included age, sex, birth weight (low, defined as < 2.5 kg), feeding practices from birth to enrollment (exclusive breastfeeding or not), and HIV exposure status. Maternal characteristics collected at baseline included age, marital status, education, HIV viral load suppression (<1000 copies/ml), type of birth attendee, malaria during pregnancy, gestational age at antenatal care (ANC) enrollment, place of infant delivery, mode of transport to nearest health facility, and maternal ART status (the timing of maternal ART initiation). Socioeconomic status (SES) index by quintiles was a composite measure derived from household income and assets.

Child mortality was defined as any child death that was verbally reported by the mother or caretaker (not a biological mother) during the follow-up period. We defined a child as an infant exposed to HIV, if it was born to a mother who had an

HIV-positive status confirmed by data from either the CHC, ANC card, maternal outpatient card, or rapid HIV test result during any study visit. Children were defined as an infant not exposed, if they were born to a mother who had an HIV-negative status and did not seroconvert during the study.

## Statistical analysis

Our outcome of interest was all-cause child mortality within the follow-up period to child age 18 months. Children who experienced the primary endpoint of the parent study (HIV infection via vertical transmission of HIV) at the baseline visit were excluded from analysis. Children who were lost to follow-up (i.e., unable to contact or locate participants for three consecutive missed appointment visits) or withdrew from the study for other reasons (i.e., refusal to continue or relocate out of the study area) and children who seroconverted during the follow-up period were censored from mortality analyses on the date recorded on the study suspension form or the last recorded study visit; their time at risk was age at their last visit. The time at risk for children who died was age at death. We calculated percentages, median values, and mortality rates per 1,000 person-years for maternal and child characteristics. We estimated the cumulative mortality probability via Kaplan–Meier curves and the log-rank test to compare differences in cumulative survival functions among infants exposed and those not exposed to HIV. We used plots of the log-log of survival functions to assess for Cox proportional hazards assumptions that were violated for most of the independent variables. For multivariate analysis, we therefore used Weibull proportional hazards and Weibull accelerated failure time (AFT) regression models with 95% confidence intervals (CIs) to identify factors associated with hazard of child death. To identify independent factors associated with child death, stepwise regression with the minimum Akaike information criterion (AIC) between the full model and the reduced model was used to identify variables to include in the final model. The strength of the association was expressed as adjusted hazard ratios and associated 95% CIs as well as failure time ratios with a 5% level of statistical significance. We used multiple imputation by chained equation (MICE) to impute missing values for variables with less than 10% missing values to avoid bias when more than 10% of the data were missing [19]. Variables with more than 10% missing data were dropped from the model. The imputed variables were infant sex, feeding practice, maternal age, marital status, socioeconomic status, malaria during pregnancy, mode of transport to health facilities, place of delivery and number of ANC visits. We conducted 20 imputations to create a set of conditional distributions for each imputed variable, which replaced the missing value with a set of plausible values that represent the uncertainty about the value to impute.

The analyses were not weighted for the complex survey design of the parent study. All the statistical analyses were conducted in Stata 16.0 (StataCorp, Lakeway Drive, Texas, USA).

## Ethical considerations

This study was reviewed and approved by the Uganda Virus Research Institute (UVRI) IRB with reference number GC/127/17/03/579 and registered with the Uganda National Council for Science and Technology (See 45 C.F.R. part 46.114; 21 C.F.R. part 56.114). All the participating mothers or caregivers provided written consent at the baseline and follow-up enrollment visits.

## Results

A total of 11,519 (68.9%) of the 16,718 infants identified were enrolled in the study, with a median age of 56 days (IQR 46–64); 50.0% were male, and 87.1% were exclusively breastfed in the first 6 months. More than half of the mothers (63.4%) were aged 15–24 years. A total of 1,723 (15.0%) children were HIV-exposed by the end of the study; of these, 63.1% of the mothers were on ART before conception, and 88.2% of the mothers with a baseline HIV viral load had a suppressed viral load. In total, 79 (0.7%) children died, 32/79 (40.5%) of whom were infants exposed to HIV (Table 1).

**Table 1. Baseline characteristics of mother–infant pairs by child mortality status (N = 11,519).**

| Characteristics | Dead, N = 79 n (%) | Alive, N = 11440 n (%) | All, N = 11519 n (%) | P value |
|---|---|---|---|---|
| **Infants** | | | | |
| **Age at baseline** (days), median (IQR) | 56 (45–65) | 56 (46–64) | 56 (46–64) | |
| **Sex** | | | | |
| Male | 37 (43.0) | 5698 (49.9) | 5735 (50.0) | |
| Female | 42 (57.0) | 5687 (50.1) | 5729 (50.0) | 0.5695 |
| **Birth weight** | | | | |
| Low (<2.5 kgs) | 1 (2.1) | 540 (7.2) | 541 (7.2) | |
| Not Low (>=2.5 kgs) | 46 (97.9) | 6949 (92.8) | 6995 (92.8) | 0.1784 |
| **Feeding practice (first 6 months)** | | | | |
| Exclusively breastfed first 6 months | 67 (85.2) | 9926 (87.1) | 9993 (87.1) | |
| Mixed/replacement feeding first 6 months | 9 (14.8) | 1378 (12.9) | 1387 (12.9) | 0.7035 |
| **HIV Exposure Status** | | | | |
| Unexposed | 47 (59.5) | 9750 (85.2) | 9797 (85.0) | |
| Exposed | 32 (40.5) | 1691 (14.8) | 1723 (15.0) | <.0001 |
| **Maternal** | | | | |
| **Age** | | | | |
| 15-24 years | 52 (69.1) | 7263 (63.4) | 7315 (63.4) | |
| 25 years and older | 27 (30.9) | 4125 (36.6) | 4152 (36.6) | 0.7063 |
| **Marital Status** | | | | |
| Living without a Partner | 26(33.8) | 2289(20.6) | 2315 (20.7) | |
| Living with a Partner | 51 (66.2) | 8829(79.4) | 8880(79.3) | 0.4220 |
| **Education** | | | | |
| Above primary | 42 (33.5) | 3532 (32.1) | 3557 (32.1) | |
| Primary and below | 54 (66.5) | 7908 (67.9) | 7962 (67.9) | 0.8816 |
| **Social Economic Status** | | | | |
| High (4th or 5th quintile) | 22 (27.4) | 4093 (36.3) | 4115 (36.3) | |
| Low (1st or 2nd quintile) | 38 (53.1) | 4914 (43.6) | 4952 (43.6) | 0.1762 |
| Middle (3rd quintile) | 17 (19.5) | 2268 (20.1) | 2285 (20.1) | 0.3051 |
| **Birth attendee** | | | | |
| Skilled | 38 (77.8) | 6904 (85.6) | 6942 (85.6) | |
| Traditional birth attendants(TBA)/Others | 11 (22.2) | 1179 (14.4) | 1190 (14.4) | 0.1248 |
| **Malaria in Preganacy** | | | | |
| Yes | 59 (77.4) | 8478 (74.4) | 8537 (74.4) | 0.8309 |
| No | 19 (22.6) | 2889 (25.6) | 2908 (25.6) | |
| **Gestation Age at start of ANC** | | | | |
| 0-3 months | 22 (31.4) | 3099 (30.4) | 3121 (30.4) | |
| 4-6 months | 43 (61.4) | 6334 (62.1) | 6377 (62.1) | 0.8651 |
| 7-9 months | 5 (7.2) | 765 (7.5) | 770 (7.5) | 0.8680 |
| **Mode of transport to health facility** | | | | |
| Walking | 51 (71.2) | 6631 (58.4) | 6682 (58.1) | 0.2421 |
| Motorized | 28 (28.8) | 4798 (41.6) | 4826 (41.9) | |
| **ART Status (HIV infected only)*** | | | | |
| Initiated preconception | 14 (51.9) | 924 (63.0) | 938 (62.8) | |
| Initated during pregnancy/Identified during pregnancy | 10 (37.0) | 417 (28.4) | 427 (28.6) | 0.2723 |
| Initiated post delivery/Identified Positive during study | 3 (11.1) | 117 (8.0) | 120 (8.0) | 0.4138 |
| Don't know | 0 (0) | 8 (0.6) | 8 (0.6) | 0.9911 |

*(Continued)*

**Table 1.** (Continued)

| Characteristics | Dead, N = 79 n (%) | Alive, N = 11440 n (%) | All, N = 11519 n (%) | *P value* |
|---|---|---|---|---|
| *Suppressed HIV Virally Load (<1000 copies/ml) at baseline(mothers living with HIV only)* | | | | |
| Suppressed | 26 (81.7) | 1283 (88.3) | 1309 (88.2) | |
| Not suppressed | 4 (18.3) | 165 (11.7) | 169 (11.8) | 0.7415 |
| **Place of delivery** | | | | |
| Public Facility | 40 (53.7) | 6905(61.4) | 6945 (61.3) | |
| Home | 17 (20.1) | 1542(13.7) | 1559 (13.8) | 0.0270 |
| Private Facility | 20 (26.2) | 2797(24.9) | 2817 (24.9) | 0.4437 |

ªMissing: Infant Sex: n = 55(0.5%); Birth weight: n = 4450 (38.6%);Feeding practice: n = 139(1.2%); Maternal age: n = 52(0.5%); Marital status: n = 320 (2.8%); Social Economic Status: n = 167 (1.5%); Viral load suppression: n = 244(2.1%); Birth attendee: n = 3387 (29.4)%; Malaria in Pregnancy: n = 74 (0.6%); Gestation Age at start of ANC: n = 1251 (10.9%); Mode of transport to health facility: n = 11 (0.1%); ART status: n = 229 (2.0%); Place of delivery: n = 194 (1.7%).

*The timing of maternal ART initiation for mothers living with HIV.

The overall child mortality rate was 5.0 per 1,000 person-years (95% CI: 4.0–6.2). HIV-exposed children and those born at home had significantly greater risks of death (p ≤ 0.0000 and p = 0.03, respectively). Mortality rates were significantly higher for infants exposed (14.2, 95% CI: 10.0–20.0) compared to those not exposed to HIV (3.5, 95% CI: 2.6–4.6); mothers not living with their partners (8.5, 95% CI: 5.8–12.4) compared to those living with a partner (4.1, 95% CI: 3.1–5.4) and infants born at home compared (7.8, 95% CI: 4.8–12.6) to those born at a public facility (4.2, 95% CI: 3.1–5.7) (Table 2).

On comparison of cumulative survival functions, survival experiences were significantly different between infants exposed and not exposed to HIV (log rank *p < 0.0001*) (Fig 1).

Compared with that of infants not exposed to HIV, the adjusted hazard ratio (aHR) of child mortality was significantly greater for infants exposed to HIV (aHR: 5.6, 95% CI: 3.4–9.2), with a median survival time reduced by 70% (Table 3). Children born to mothers aged 15--24 years had a significantly greater hazard (aHR: 1.8, 95% CI: 1.1--2.9) than those born to mothers aged ≥25 years, with the median survival time reduced by 30%. There was a significantly greater hazard ratio (aHR: 1.8, 95% CI: 1.1--2.9) for mothers living without a partner than those living with a partner, with the median survival time ratio reduced by 30%. Home-delivered babies had a significantly greater hazard rate (aHR: 2.3; 95% CI: 1.3–4.0) than did those delivered in a public facility, with a reduced median survival time of 20%.

The overall Weibull model shape parameter of p = 1.37 suggested that the hazard of child mortality generally increased with time.

## Discussion

The overall mortality rate of 5.0 deaths per 1,000 person-years observed in the study cohort (with follow-up to 18 months) was six times lower than the 32.9 deaths per 1,000 person-years reported in a 2009–2011 cohort of children from rural Uganda over a median follow-up of 2 years [20]. Of note,in 2011 13.8% of children under five years of age were under weight with increased vulnerability to dieases; the infant and under-five mortality rates were 54/1000 and 90/1000 live births respectively with HIV prevalence at 7.3% [21,22]. The lower observed rate in this analysis could reflect that all infants in the study were engaged in health care regularly for study visits at the clinics where they could receive health education, vaccinations and child nutrition monitoring, which improved the quality of child health services and thus reduced the number of preventable child deaths [23]. Furthermore, infants exposed to HIV and their mothers had access to prevention of vertical transmission of HIV interventions, including maternal ART; the use of cotrimoxazole prophylaxis to prevent opportunistic infections, malaria, and upper respiratory infections; and the promotion of breastfeeding following the guidance of the World Health Organization (WHO) and Ministry of Health (MOH), which recommends a longer duration of breastfeeding instead of replacement feeding for infants exposed to HIV [24–26].

**Table 2. Child mortality rate per 1,000 person-years during the 18-month follow-up period.**

| Characteristic | Deaths | Number of Infants | Person years of infant age | MR per 1000 person years | 95%CI | p values |
|---|---|---|---|---|---|---|
| *Infants* | | | | | | |
| Overall | 79 | 11519 | 15870.5 | 5 | [4.0–6.2] | |
| *Sex* | | | | | | |
| Male | 37 | 5735 | 7894.6 | 4.7 | [3.4–6.5] | |
| Female | 42 | 5730 | 7905.4 | 5.3 | [3.9–7.2] | 0.5781 |
| *Feeding practice (first 6 months)* | | | | | | |
| Mixed/replacement feeding first 6 months | 9 | 1387 | 1921.7 | 4.7 | [2.4–9.0] | |
| Exclusively breastfed first 6 months | 67 | 9993 | 13760.2 | 4.9 | [3.8–6.1] | 0.7110 |
| *HIV exposure status* | | | | | | |
| Unexposed | 47 | 9797 | 13613.2 | 3.5 | [2.6–4.6] | |
| Exposed | 32 | 1723 | 2257.3 | 14.2 | [10.0–20.0] | <.0001 |
| *Maternal* | | | | | | |
| *Age* | | | | | | |
| 15-24 years | 52 | 7316 | 10068.2 | 5.2 | [3.9–6.8] | |
| 25 years and older | 27 | 4152 | 5732.1 | 4.7 | [3.2–6.9] | 0.6978 |
| *Marital Status* | | | | | | |
| Living without a partner | 26 | 2315 | 3068.4 | 8.5 | [5.8–12.4] | |
| Living with a partner | 51 | 8880 | 12380.2 | 4.1 | [3.1–5.4] | 0.0028 |
| *Education* | | | | | | |
| Above Primary Level | 25 | 3552 | 4919.3 | 5.1 | [3.4–7.5] | |
| None/Primary/Don't Know | 54 | 7741 | 10951.3 | 4.9 | [3.8–6.4] | 0.9007 |
| *Social Economic Status* | | | | | | |
| High (4th or 5th quintile) | 22 | 4115 | 5646.7 | 3.9 | [2.6–5.9] | |
| Low (1st or 2nd quintile) | 38 | 4952 | 6832.5 | 5.6 | [4.0–7.6] | 0.1840 |
| Middle (3rd quintile) | 17 | 2285 | 3162.7 | 5.4 | [3.3–8.6] | 0.3190 |
| *Birth attendee* | | | | | | |
| Skilled | 38 | 6942 | 9526.0 | 4.0 | [2.9–5.5] | |
| Traditional birth attendant (TBA)/Other | 11 | 1190 | 1645.3 | 6.7 | [3.7–12.1] | 0.1315 |
| *Malaria in preganacy* | | | | | | |
| No | 59 | 8537 | 11769.9 | 5.0 | [3.9–6.5] | |
| Yes | 19 | 2908 | 4001.6 | 4.7 | [3.0–7.4] | 0.8370 |
| *Gestation Age at start of ANC* | | | | | | |
| 0-3 months | 22 | 3121 | 4263.8 | 5.2 | [3.4–7.8] | |
| 4-6 months | 43 | 6377 | 8813.7 | 4.9 | [3.6–6.6] | 0.8309 |
| 7-9 months | 5 | 770 | 1058.7 | 4.7 | [1.9–11.3] | 0.8582 |
| *Mode of transport to Health Facility* | | | | | | |
| Motorize | 28 | 4826 | 6640.7 | 4.2 | [2.9–6.1] | |
| Walking | 51 | 6682 | 9214.5 | 5.5 | [4.2–7.3] | 0.2474 |
| *ART Status (HIV infected only)* | | | | | | |
| Initiated preconception | 14 | 938 | 1256.2 | 11.1 | [6.6–18.8] | |
| Initated during pregnancy/Identified during pregnancy | 10 | 427 | 549.8 | 18.2 | [9.7–33.8] | 0.2368 |
| Initiated post delivery/Identified Positive during study | 3 | 120 | 143.1 | 21 | [6.7–65.0] | 0.3205 |
| *Suppressed HIV Viral Load (<1000 copies/ml) at baseline(mothers living with HIV only)* | | | | | | |
| Suppressed | 26 | 1310 | 1737.4 | 15.0 | [10.2–22.0] | |
| Not suppressed | 4 | 169 | 214.2 | 18.7 | [7.0–49.8] | 0.6808 |

*(Continued)*

**Table 2.** (Continued)

| Characteristic | Deaths | Number of Infants | Person years of infant age | MR per 1000 person years | 95%CI | p values |
|---|---|---|---|---|---|---|
| **Place of delivery** | | | | | | |
| Public Facility | 40 | 6945 | 9557.3 | 4.2 | [3.1–5.7] | |
| Home | 17 | 1559 | 2164.1 | 7.8 | [4.8–12.6] | 0.0296 |
| Private Facility | 20 | 2817 | 3885.8 | 5.1 | [3.3–7.9] | 0.4501 |

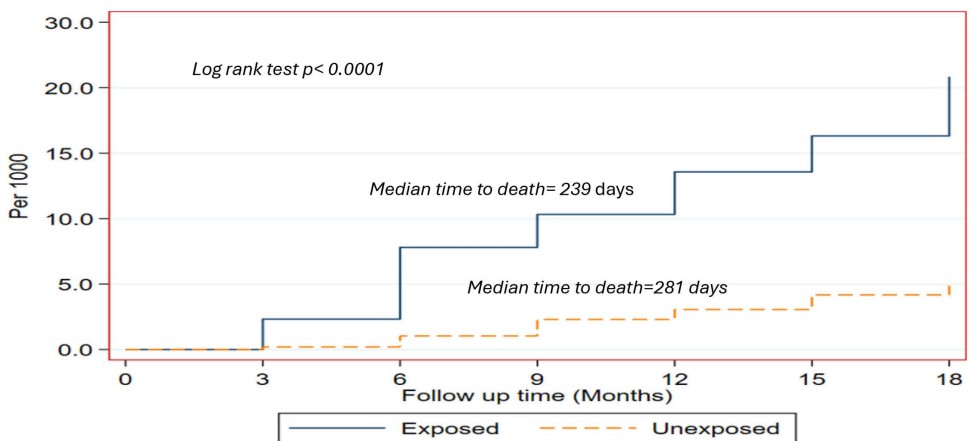

y -axis scaled to out of 1000 infants.

| | Exposed | | Unexposed | | Total | |
|---|---|---|---|---|---|---|
| Interval (Months) | Deaths | Cum. Failure | Deaths | Cum. Failure | Deaths | Cum. Failure |
| 1 | 0 | 0.0000 | 0 | 0.0000 | 0 | 0.0000 |
| 3 | 4 | 0.0023 | 2 | 0.0002 | 6 | 0.0005 |
| 6 | 9 | 0.0078 | 8 | 0.0010 | 17 | 0.0020 |
| 9 | 4 | 0.0103 | 12 | 0.0023 | 16 | 0.0035 |
| 12 | 5 | 0.0136 | 7 | 0.0031 | 12 | 0.0046 |
| 15 | 4 | 0.0163 | 10 | 0.0042 | 14 | 0.0059 |
| 18 | 6 | 0.0208 | 8 | 0.0051 | 14 | 0.0074 |

**Fig 1. Kaplan–Meier cumulative survival curve for infants exposed and not exposed to HIV.**

In this study, an approximately fourfold greater child mortality rate was observed among infants exposed than among those not exposed to HIV. These findings are consistent with the available literature showing that infants exposed to HIV have higher mortality rates than those not exposed to HIV do, ranging from 4.0–13.6% [8,27,28]. This could be due to the infant feeding options available, such as exclusive breastfeeding or no breastfeeding, which poses different dilemmas. For example, longer breastfeeding may increase the vertical transmission of HIV and thus HIV-associated mortality, whereas shorter or no breastfeeding may expose the uninfected child to common illnesses such as pneumonia that lead to mortality [29,30]. Notably, there was no significant difference in mortality rates between exclusively breastfed infants and those receiving mixed or replacement feeding in the first 6 months, although studies have documented greater mortality rates among infants exposed than those not exposed to HIV with similar feeding patterns [8,31,32]. Other studies have shown that even after the introduction of prevention of vertical transmission of HIV policies and programs, infants exposed are at increased risk of severe bacterial infections, impaired cognitive function and death compared with those not exposed

 

**Table 3. Crude and adjusted times and hazard ratios for child mortality.**

| | Crude Time Ratio [95%CI] | Adj. Time Ratio [95%CI] | Crude Weibull HR [95%CI] | Adj.Weibull HR [95%CI] |
|---|---|---|---|---|
| **Cohort** | | | | |
| HIV Unexposed | **1** | | **1** | |
| HIV Exposed | **0.4 [0.2–0.5]** | **0.3 [0.2–0.4]** | **4.1 [2.6–6.6]** | **5.6 [3.4–9.2]** |
| *Mother's Age* | | | | |
| 25+years | 1 | | 1 | |
| 15-24 years | 0.9 [0.7–1.3]] | **0.7 [0.5–0.9]** | 1.1 [0.7–1.7] | **1.8 [1.1–2.9]** |
| *Infant Sex* | | | | |
| Male | 1 | | 1 | |
| Female | 0.9 [0.7–1.3] | 0.9 [0.7–1.2] | 1.1 [0.7–1.7] | 1.2 [0.7–1.8] |
| *Family Social Economic Status* | | | | |
| High (4th or 5th quintile) | 1 | | 1 | |
| Low (1st or 2nd quintile) | 0.8 [0.5–1.2] | 0.7 [0.4–1.0] | 1.4 [0.8–2.4] | 1.7 [1.0–3.0] |
| Middle (3rd quintile) | 0.8 [0.5–1.3] | 0.7 [0.4–1.1] | 1.4 [0.7–2.5] | 1.6 [0.8–3.0] |
| *Mother's Marital Status* | | | | |
| Living with a partner | 1 | | 1 | |
| Living without a partner | **0.6 [0.4–0.9]** | **0.7 [0.5–0.9]** | **2.0 [1.2–3.4]** | **1.8 [1.1–2.9]** |
| **Mother's Education** | | | | |
| Above Primary level | 1 | | 1 | |
| None/Primary | 1.0 [0.7–1.4] | | 1.0 [0.6–1.6] | |
| **Malaria in Pregnancy** | | | | |
| No | 1 | | 1 | |
| Yes | 1.0 [0.7–1.5] | | 0.9 [0.6–1.6] | |
| **Birth attendee** | | | | |
| Skilled | 1 | | 1 | |
| Traditional birth attendants (TBA)/Other | 0.7 [0.4–1.1] | | 1.9 [1.1–3.4] | |
| **Mode of transport to health facility** | | | | |
| Motorized | 1 | | 1 | |
| Walking | 0.8 [0.6–1.2] | 0.8 [0.5–1.1] | 1.3 [0.7–2.2] | 1.5 [0.9–2.3] |
| **Breastfeeding pattern** | | | | |
| Exclusive 1st 6 months | 1 | | 1 | |
| Mixed/Replacement | 1.0 [0.7–1.8] | 1.0 [0.6–1.7] | 1.0 [0.4–2.0] | 1.0 [0.5–1.9] |
| **Attended 4 or more ANC visits** | | | | |
| No | 1 | | 1 | |
| Yes | 1.1 [0.7–1.8] | | 1.1 [0.4–3.0] | |
| **ART Status (HIV infected only)** | | | | |
| Preconception | 1 | | 1 | |
| During pregnancy | 0.7 [0.4–1.3] | | 1.6 [0.7–3.5] | |
| Post delivery | 0.6 [0.2–1.7] | | 1.7 [0.5–5.7] | |

*(Continued)*

**Table 3.** (Continued)

| | Crude Time Ratio [95%CI] | Adj. Time Ratio [95%CI] | Crude Weibull HR [95%CI] | Adj.Weibull HR [95%CI] |
|---|---|---|---|---|
| **Mother baseline Viral load non suppression(only Exposed)** | | | | |
| No | 1 | | 1 | |
| Yes | 0.8 [0.3–2.0] | | 1.4 [0.4–3.6] | |
| **Place of Delivery** | | | | |
| Public Facility | 1 | | 1 | |
| Private Facility | 0.9 [0.6–1.3] | 0.8 [0.5–1.1] | 1.2 [0.7–2.1] | 1.4 [0.8–2.5] |
| Home | **0.6 [0.4–0.9]** | **0.8 [0.4–0.8]** | **1.9 [1.1–3.5]** | **2.3 [1.3–4.0]** |

The effective sample size was 11,514 and the bolded figures represent the significant variables.

to HIV. While the causes of these poor outcomes are multifactorial, studies suggest that this could be attributed to the impaired innate and adaptive immune responses observed in infants exposed to HIV, secondary to ART exposure and maternal HIV infection [33,34].

A mother's age of less than 25 years was a significant risk factor for child mortality, which is consistent with several studies that have shown associations between young maternal age and poor child survival [20,35–39]. This is especially relevant for Uganda, with a teenage pregnancy rate of 25% and a high percentage of women under the age of 25 [40]. Young maternal age is associated with child mortality risk factors such as low birth weights, preterm deliveries, limited exposure to reproductive health and low maternal health-seeking behaviors, especially postnatal care [40–43]. Low birth weight is a known risk factor for infant mortality and influences the health outcomes of newborns [44,45]. In this analysis, the absence of infant birthweight data precluded its inclusion in the analytical models. With the estimated unmet need for family planning at 30.4% among women aged 15--19 years in Uganda, strategies such as additional socioeconomic and educational support and meeting the family planning needs of young mothers may further reduce child mortality rates [35,40,46–48].

In this analysis, both mothers living without a partner and those delivering an infant at home significantly increased the risk of child mortality. Nabongo et al., in their cohort study of children in rural Uganda, reported that the child's birthplace, which was not a health facility, was associated with child mortality [20]. These factors could be markers of a lower socioeconomic status of the household [49]. Studies have revealed that low economic status is associated with infant mortality because it affects access to healthcare, nutrition, and social services [11,50,51]. The quality of health care received by the infant and mother is often determined by the place of delivery, with infants delivered at home not receiving the full package of postnatal care. Sarmistha provides new evidence that institutional delivery can significantly lower child mortality risk because it ensures effective and timely access to modern diagnostics and medical treatments to save lives [52]. While public health and treatment programs are in place to address the availability of healthcare services, interventions that enhance the economic well-being of the lower socioeconomic group might also have positive downstream impacts on child mortality. There were at least five limitations in this study. First, the total number of infant deaths identified was small, limiting the ability to perform regression analyses. Second, MIPs were recruited from child immunization clinics, creating selection bias in the sample. While immunization rates in Uganda are high, MIPs not engaging in early childhood care or engaging outside the enrollment age would not have been screened for participation in the study. The selection approach likely excluded children who were not immunized and were at high risk of mortality from vaccine-preventable diseases [53]. Therefore, the study population might have represented a

healthier subset of the broader community, which potentially underestimated the true burden of mortality in the general population. Third, the study was not able to actively follow-up MIPs that were transferred to areas outside of the 152 study sites. As a result, transient MIPs may have been lost to follow-up. Fourth, selection bias could also have been introduced because of the study eligibility criteria, which excluded MIP due to severe illness of either the infant or the mother, mothers who were unable to give informed consent and those who refused the infant's HIV antibody test at baseline. Finally, there were significant missing data for some of the critical variables, such as birth weight, due to a lack of source documents.

Mortality among infants exposed to HIV was significantly greater than that among those not exposed to HIV. Several factors, including HIV exposure, younger maternal age, the absence of a male partner, and home delivery of the child, increase the risk of mortality. Efforts to reduce child mortality could consider screening for these factors to be able to provide interventions during pregnancy and through the postpartum period for mothers whose children are at increased risk. Future studies are needed to explore how these factors drive mortality in this age group.

## Supporting information

**S1 Dataset. Mortality dataset used in analysis.**
(XLS)

**S2 Data. Dictionary.** Varible descriptions within the mortality dataset.
(DOCX)

## Acknowledgments

The authors thank the prevention of vertical transmission of HIV impact evaluation study staff, study participants and health facility teams for their time and effort. All the authors contributed to the study design, study conduct, data analysis and interpretation, or manuscript writing.

The findings and conclusions in this manuscript are those of the author(s) and do not necessarily represent the official position of the funding agencies.

## Author contributions

**Conceptualization:** Samuel Sendagala, Linda Nabitaka, Fred Nalugoda, Margaret Achom Okwero, Hilda T Asiimwe, Phoebe Namukanja, Bernadette Ng'eno, Augustina Delaney.

**Data curation:** Rose Bosa Nakityo, Ivan Lukabwe.

**Formal analysis:** Samuel Sendagala, Rose Bosa Nakityo, Fredrick Makumbi, Tom Lutalo, Ivan Lukabwe, Augustina Delaney.

**Investigation:** Linda Nabitaka, Fred Nalugoda, Jingo Kasule, Bernadette Ng'eno.

**Methodology:** Samuel Sendagala, Rose Bosa Nakityo, Fredrick Makumbi, Tom Lutalo, Fred Nalugoda, Emily Namara-Lugolobi, Margaret Achom Okwero, Hilda T Asiimwe, Phoebe Namukanja, Emilio Dirlikov, Augustina Delaney.

**Project administration:** Linda Nabitaka, Fred Nalugoda, Jingo Kasule, Phoebe Namukanja.

**Supervision:** Linda Nabitaka, Jingo Kasule, Phoebe Namukanja, Emilio Dirlikov.

**Writing – original draft:** Samuel Sendagala.

**Writing – review & editing:** Samuel Sendagala, Rose Bosa Nakityo, Fredrick Makumbi, Tom Lutalo, Linda Nabitaka, Fred Nalugoda, Ivan Lukabwe, Jingo Kasule, Emily Namara-Lugolobi, Margaret Achom Okwero, Hilda T Asiimwe, Phoebe Namukanja, Bernadette Ng'eno, Emilio Dirlikov, Augustina Delaney.

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
