## [Decision Letter · Decision Letter 0]

6 Dec 2024

PONE-D-24-43587Mortality rates in a cohort of infants attending immunization clinics in Uganda (2017–2019)PLOS ONE

Dear Dr. Sendagala,

Thank you for submitting your manuscript to PLOS ONE. After careful consideration, we feel that it has merit but does not fully meet PLOS ONE’s publication criteria as it currently stands. Therefore, we invite you to submit a revised version of the manuscript that addresses the points raised during the review process.

The authors present a review of infant/child mortality rates by HIV-exposure status of attendees at immunization clinics in Uganda in the context of a mature maternal HIV treatment programme. Mortality rates were significantly higher in infants/children of younger mother who were HIV exposed.

The manuscript is well-written but requires revision in preparation for publication.

1. Language. As noted by Reviewer 2, there is a move towards patient-centred language in the HIV field. Women living with HIV, prevention of vertical transmission of HIV etc. Please adjust all language in the title, abstract and manuscript.

2. Please provide more detail with respect to the Methodology and Results as described by both reviewers.

3. Please consider review of the limitations in light of the reviewer comments.

4. In addition, there are further queries and clarifications noted by the reviewers which will strengthen the paper.

We look forward to receiving your revised manuscript.

Kind regards,

Emma K. Kalk

Academic Editor

PLOS ONE

Journal Requirements:

2. Thank you for stating the following financial disclosure: “This research has been supported by the President's Emergency Plan for AIDS Relief (PEPFAR) through the Centers for Disease Control and Prevention (CDC) under the terms of cooperative agreement number 5U2G GH000817-02.”

3. We note that you have indicated that there are restrictions to data sharing for this study. PLOS only allows data to be available upon request if there are legal or ethical restrictions on sharing data publicly. For more information on unacceptable data access restrictions, please see http://journals.plos.org/plosone/s/data-availability#loc-unacceptable-data-access-restrictions. Before we proceed with your manuscript, please address the following prompts: a) If there are ethical or legal restrictions on sharing a de-identified data set, please explain them in detail (e.g., data contain potentially identifying or sensitive patient information, data are owned by a third-party organization, etc.) and who has imposed them (e.g., a Research Ethics Committee or Institutional Review Board, etc.). Please also provide contact information for a data access committee, ethics committee, or other institutional body to which data requests may be sent. b) If there are no restrictions, please upload the minimal anonymized data set necessary to replicate your study findings to a stable, public repository and provide us with the relevant URLs, DOIs, or accession numbers. For a list of recommended repositories, please see https://journals.plos.org/plosone/s/recommended-repositories. You also have the option of uploading the data as Supporting Information files, but we would recommend depositing data directly to a data repository if possible. We will update your Data Availability statement on your behalf to reflect the information you provide.

Additional Editor Comments:

The authors present a review of infant/child mortality rates by HIV-exposure status of attendees at immunization clinics in Uganda in the context of a mature maternal HIV treatment programme. Mortality rates were significantly higher in infants/children of younger mother who were HIV exposed.

The manuscript is well-written but requires revision in preparation for publication.

1. Language. As noted by Reviewer 2, there is a move towards patient-centred language in the HIV field. Women living with HIV, prevention of vertical transmission of HIV etc. Please adjust all language in the title, abstract and manuscript.

2. Please provide more detail with respect to the Methodology and Results as described by both reviewers.

3. Please consider review of the limitations in light of the reviewer comments.

4. In addition, there are further queries and clarifications noted by the reviewers which will strengthen the paper.

Reviewers' comments:

Reviewer's Responses to Questions

**Comments to the Author**

1. Is the manuscript technically sound, and do the data support the conclusions?

Reviewer #1: Partly

Reviewer #2: Partly

2. Has the statistical analysis been performed appropriately and rigorously? 

Reviewer #1: No

Reviewer #2: I Don't Know

3. Have the authors made all data underlying the findings in their manuscript fully available?

Reviewer #1: Yes

Reviewer #2: No

4. Is the manuscript presented in an intelligible fashion and written in standard English?

Reviewer #1: Yes

Reviewer #2: Yes

5. Review Comments to the Author

Reviewer #1: The manuscript covers an important topic, but certain methodological and interpretative issues need to be addressed to enhance clarity and rigor. Including additional significance testing, presenting absolute data on events, and expanding the discussion of unexpected findings would greatly strengthen the paper.

Here a list of suggestions:

- Abstract

-- Methods: The abstract should clarify where the study took place and the inclusion criteria for the participants. For example, were these women who brought their children to immunization clinics? This needs to be explicitly mentioned.

Additionally, the abstract should specify which variables were studied.

-- Conclusions: The conclusion about home delivery does not seem to be the most significant finding and could be reconsidered.

- Introduction

-- Confidence Intervals: It is unnecessary to include confidence intervals for descriptive data in this section (e.g., line 47). These figures are meant to provide context, and whether the estimate is 2.6 million or 3.1 million does not affect the overall argument. The same applies to line 53.

-- Option B+ Reference: On lines 69–71, is there a reference to support the introduction of Option B+ in Uganda? If not, this should be included.

-- HIV Prevalence: The introduction should explicitly include data on HIV prevalence in Uganda. The phrase "a country with a mature generalized HIV epidemic" is vague and should be clarified with specific prevalence data. How many people are affected? Why is it important to study HIV and infant mortality? These points should not be taken for granted, as the introduction must justify every aspect of the study design.

- Methods

-- S1 Text Reference: The reference to S1 Text is unclear. Is it a paper under evaluation for publication? If so, this should be explicitly stated. Citing a Word document is not appropriate for academic referencing. If the text is published, cite the published version. If it is under evaluation, clarify its status and provide more detail about its content.

-- Study Design and Sampling: The inclusion criteria and sampling methods need to be explained more clearly. For example, when and how were the women recruited? Were all women included? The methodology should provide a detailed explanation of how the study was conducted.

- Results

-- Table 1: Consider including tests of significance to identify differences between the samples of deceased and surviving participants.

-- Table 2: Similarly, tests of significance could be added here to improve the robustness of the analysis.

-- Key Findings: The paper primarily discusses "when" events occur (e.g., which groups have shorter survival times). However, it is unclear whether it also addresses "how many" events occur in each group (e.g., the absolute number or proportion of deaths or transmissions). To strengthen the results, I suggest additional analyses:

--- Significance testing: Compare the proportions of events (e.g., transmissions or deaths) between groups with different characteristics using chi-squared tests for categorical variables or t-tests/Mann-Whitney U tests for continuous variables.

--- Cumulative risk analysis: Include Kaplan-Meier survival curves and log-rank tests to analyze both the timing and the number of events.

--- Absolute data: Present the absolute number of events (e.g., transmissions or deaths) for each group to complement the findings on timing.

--- These additional analyses would address the critical question: do certain characteristics reduce the overall risk of death/transmission, or do they merely delay it?

- Discussion

-- Comparison with Older Data: Comparing the study sample with rural data from 15 years ago may not be the most appropriate choice. Are there more recent studies or datasets, such as World Bank data, that could provide a better comparison? If you decide to compare with the older study, include context about the previous cohort, such as HIV prevalence, maternal nutritional conditions, and other relevant factors.

-- Time in Care and Viral Load: It is surprising that time in care and viral load were not associated with the outcomes. This finding is inconsistent with existing evidence, which shows that longer time in care is associated with reduced neonatal infections and improved survival rates. If your findings diverge from this, it is essential to discuss why this might be the case.

-- Explore better limitations of the study related to selection bias. Another limitation of the study is the potential selection bias introduced by recruiting only mother-infant pairs attending immunization clinics. This approach likely excluded children who were not immunized and who may be at higher risk of mortality from vaccine-preventable diseases. As a result, the study population may represent a healthier subset of the broader community, potentially underestimating the true burden of mortality and transmission in the general population.

Reviewer #2: This manuscript describes child mortality rates, up to age 18 months, and associated risk factors, in Uganda, with a particular focus on comparing mortality rates by child HIV exposure status. It provides a valuable contribution to the field of research, particularly as the study period (2017 – 2018) coincided with the Universal ART era - a time marked by increased maternal access to ART and a reduction in vertical HIV transmission rates. Please consider addressing the following comments to help strengthen this manuscript.

Comments to authors:

General

1. General - The comparison of child mortality rates by HIV exposure status appears to be a main focus of this study, however this was not listed as one of the aims. Consider bringing this into the aims.

2. General – Language. Authors should be careful to use non stigmatizing person-first language. Consider infants who are exposed to HIV and infants unexposed to HIV instead of HIV exposed infants and HIV unexposed infants. Consider HIV vertical transmission prevention (VTP) instead of PMTCT.

3. General – Authors should consider carefully reviewing the manuscript for inconsistencies in spaces between words.

Abstract

4. Abstract – Line 31. Authors should consider adding that they used multivariate regression.

Introduction

5. Introduction - Line 53: 42 deaths (29-60) per 1,000 live births in 2021. Please specify if the (29-60) is a 95% CI.

6. Introduction – Line 65. Considering including and referencing Evans C, Chasekwa B, Ntozini R, Majo FD, Mutasa K, Tavengwa N, Mutasa B, Mbuya MNN, Smith LE, Stoltzfus RJ, Moulton LH, Humphrey JH, Prendergast AJ; Sanitation Hygiene Infant Nutrition Efficacy (SHINE) Trial Team. Mortality, Human Immunodeficiency Virus (HIV) Transmission, and Growth in Children Exposed to HIV in Rural Zimbabwe. Clin Infect Dis. 2021 Feb 16;72(4):586-594. doi: 10.1093/cid/ciaa076. PMID: 31974572; PMCID: PMC7884806.

7. Introduction – Check consistency in number of decimal places used throughout.

Methods

8. Methods – Which participants were excluded from the study?

9. Were mothers and children HEI tested for HIV at every visit? Is it possible that they weren’t tested and that there could have been misclassification of children HEI and HUI? If so, this should be discussed under limitations. Children HEI that die may have been living with HIV at the time of death. Children HUI also need to have been tested to be sure that they are not living with HIV. Although less common, children may acquire HIV horizontally.

10. Methods – Line 96. How was maternal ART status defined?

11. Methods – Line 115. Should “Wilcoxon log-rank test” just be log rank test?

12. Methods – Line 116. The authors state that they compared the differences in cumulative survival functions among HEIs and HUIs, however, these results were not shown in the results section.

13. Methods – Line 119. Authors should consider clarifying that they did Weibull proportional hazards regression and Weibull accelerated failure time regression (two different models).

14. Methods – Line 124. The authors should justify only using multiple imputation for variables with <10% missing values. What was done with variables that have >10% missing values and how did this affect your sample size for analyses?

Results

15. Results – Table 1. Authors should consider positioning HIV related variables (maternal viral load and maternal ART) together in the table.

16. Results – Table 1. For several of the variables (e.g. Feeding practice, maternal age, marital status) the reported proportions need to be revised as they are slightly off, e.g. the proportion of all mothers living with a partner should be 79.3% instead of 78.2%.

17. Under ART status, for the proportion of mothers who initiated during pregnancy, is this only among women who were diagnosed with HIV before pregnancy? Authors to consider a footnote to clarify.

18. Line 155 – Should this refer to cumulative probability or mortality rate? Consider ensuring the sentence doesn’t fall between Tables 1 and 2.

19. Table 2 – What does unweighted number refer to?

20. Table 2 - Consider positioning overall mortality above and separate from sex.

21. Table 3 – Include time ratios in the table caption.

22. Table 3 – What were the sample sizes for the adjusted models?

23. Table 3 – Consider a footnote for the table explaining why some of the estimates are bolded.

24. Line 169 – 170 – This figure refers to figure 1, but p=1.37 is not shown in the figure. Consider going into more detail about the results in Figure 1.

25. Figure 1 - It may be helpful to have a footnote explaining that the y axis has been scaled to be out of 1000.

26. Figure 1 – There are three smaller graphs. Two are stratified by mothers age and gender respectively and the third, showing cumulative probability of infant mortality/1000, is not stratified. It is unclear whether the third graph is meant to be a Kaplan Meier curve. Additionally, these three graphs were not mentioned in the methods section. If the authors choose to include the three small graphs, they need to be referred to in the methods section and more clearly explained in the figure caption.

Discussion

27. Discussion line 193 - “4.0-13.6”, are these deaths per 1000 person years?

28. Line 203 – “adoptive” should be “adaptive”

29. Discussion line 235 – These exclusion criteria need to be included in the methods section.

6. PLOS authors have the option to publish the peer review history of their article (what does this mean? ). If published, this will include your full peer review and any attached files.

**Do you want your identity to be public for this peer review?** For information about this choice, including consent withdrawal, please see our Privacy Policy .

Reviewer #1: **Yes: ** Fausto Ciccacci

Reviewer #2: No

---

## [Author Response · Author response to Decision Letter 0]

7 Mar 2025

Dear All,

Thank you for the comprehensive and constructive feedback you have given on our manuscript. All the support to make the manuscript better are much appreciated.

Below are my responses to the comments which have also been reflected in the resubmitted manuscript.

Sam

Journal Requirements:

Response: Noted

2. Thank you for stating the following financial disclosure: “This research has been supported by the President's Emergency Plan for AIDS Relief (PEPFAR) through the Centers for Disease Control and Prevention (CDC) under the terms of cooperative agreement number 5U2G GH000817-02.”

Response: Edited to reflect the amended role of the funder

3. We note that you have indicated that there are restrictions to data sharing for this study. PLOS only allows data to be available upon request if there are legal or ethical restrictions on sharing data publicly. For more information on unacceptable data access restrictions, please see http://journals.plos.org/plosone/s/data-availability#loc-unacceptable-data-access-restrictions. Before we proceed with your manuscript, please address the following prompts:

b) If there are no restrictions, please upload the minimal anonymized data set necessary to replicate your study findings to a stable, public repository and provide us with the relevant URLs, DOIs, or accession numbers. For a list of recommended repositories, please see https://journals.plos.org/plosone/s/recommended-repositories.

You also have the option of uploading the data as Supporting Information files, but we would recommend depositing data directly to a data repository if possible. We will update your Data Availability statement on your behalf to reflect the information you provide.

Response: The minimal anonymized data set has been shared.

Response: Full name of the IRB of record and the consent statement have been included under “Ethical considerations”

Additional Editor Comments:

The authors present a review of infant/child mortality rates by HIV-exposure status of attendees at immunization clinics in Uganda in the context of a mature maternal HIV treatment programme. Mortality rates were significantly higher in infants/children of younger mother who were HIV exposed.

The manuscript is well-written but requires revision in preparation for publication.

1. Language. As noted by Reviewer 2, there is a move towards patient-centered language in the HIV field. Women living with HIV, prevention of vertical transmission of HIV etc. Please adjust all language in the title, abstract and manuscript.

Response: The manuscript has been reviewed and the language edited to match the patient-centered language.

2. Please provide more detail with respect to the Methodology and Results as described by both reviewers.

Response: Edited to provide the details.

3. Please consider review of the limitations in light of the reviewer comments.

Response: Reviewed and edited to address the reviewer’s comments.

4. In addition, there are further queries and clarifications noted by the reviewers which will strengthen the paper.

Reviewers' comments:

Reviewer's Responses to Questions

Comments to the Author

1. Is the manuscript technically sound, and do the data support the conclusions?

Reviewer #1: Partly

Reviewer #2: Partly

2. Has the statistical analysis been performed appropriately and rigorously?

Reviewer #1: No

Reviewer #2: I Don't Know

3. Have the authors made all data underlying the findings in their manuscript fully available?

The PLOS Data policy requires authors to make all data underlying the findings described in their manuscript fully available without restriction, with rare exception (please refer to the Data Availability Statement in the manuscript PDF file). The data should be provided as part of the manuscript or its supporting information or deposited to a public repository. For example, in addition to summary statistics, the data points behind means, medians and variance measures should be available. If there are restrictions on publicly sharing data—e.g. participant privacy or use of data from a third party—those must be specified.

Reviewer #1: Yes

Reviewer #2: No

4. Is the manuscript presented in an intelligible fashion and written in standard English?

Reviewer #1: Yes

Reviewer #2: Yes

5. Review Comments to the Author

Reviewer #1:

The manuscript covers an important topic, but certain methodological and interpretative issues need to be addressed to enhance clarity and rigor. Including additional significance testing, presenting absolute data on events, and expanding the discussion of unexpected findings would greatly strengthen the paper.

Response: Thank you for your time to comprehensively review our manuscript and the constructive feedback you have given. Below are my responses to the comments which have also been reflected in the resubmitted manuscript.

Here a list of suggestions:

Abstract

-- Methods: The abstract should clarify where the study took place and the inclusion criteria for the participants. For example, were these women who brought their children to immunization clinics? This needs to be explicitly mentioned.

Response: Edited and included.

Additionally, the abstract should specify which variables were studied.

Response: Edited and included.

-- Conclusions: The conclusion about home delivery does not seem to be the most significant finding and could be reconsidered.

Response: Reconsidered and Edited

Introduction

-- Confidence Intervals: It is unnecessary to include confidence intervals for descriptive data in this section (e.g., line 47). These figures are meant to provide context, and whether the estimate is 2.6 million or 3.1 million does not affect the overall argument. The same applies to line 53.

Response: Noted and Edited

-- Option B+ Reference: On lines 69–71, is there a reference to support the introduction of Option B+ in Uganda? If not, this should be included.

Response: Reference added

-- HIV Prevalence: The introduction should explicitly include data on HIV prevalence in Uganda. The phrase "a country with a mature generalized HIV epidemic" is vague and should be clarified with specific prevalence data. How many people are affected? Why is it important to study HIV and infant mortality? These points should not be taken for granted, as the introduction must justify every aspect of the study design.

Response: Noted and the content edited to address the comments.

Methods

-- S1 Text Reference: The reference to S1 Text is unclear. Is it a paper under evaluation for publication? If so, this should be explicitly stated. Citing a Word document is not appropriate for academic referencing. If the text is published, cite the published version. If it is under evaluation, clarify its status and provide more detail about its content.

Response: The paper is under evaluation for the publication. With your guidance, we have dropped the reference and edited the methods section to provide the methodology details.

-- Study Design and Sampling: The inclusion criteria and sampling methods need to be explained more clearly. For example, when and how were the women recruited? Were all women included? The methodology should provide a detailed explanation of how the study was conducted.

Response: We have edited the methods section to provide the missing details.

Results

-- Table 1: Consider including tests of significance to identify differences between the samples of deceased and surviving participants.

Response: Edited and included.

-- Table 2: Similarly, tests of significance could be added here to improve the robustness of the analysis.

Response: Done

-- Key Findings: The paper primarily discusses "when" events occur (e.g., which groups have shorter survival times). However, it is unclear whether it also addresses "how many" events occur in each group (e.g., the absolute number or proportion of deaths or transmissions). To strengthen the results, I suggest additional analyses:

Response: Additional analysis done, and a table has been added to figure 1

--- Significance testing: Compare the proportions of events (e.g., transmissions or deaths) between groups with different characteristics using chi-squared tests for categorical variables or t-tests/Mann-Whitney U tests for continuous variables.

Response: Additional analysis done, and p-values added to the tables 1 and 2

--- Cumulative risk analysis: Include Kaplan-Meier survival curves and log-rank tests to analyze both the timing and the number of events.

Response: Figure 1 revised to include the suggestions.

--- Absolute data: Present the absolute number of events (e.g., transmissions or deaths) for each group to complement the findings on timing.

Response: Done and included in figure 1

--- These additional analyses would address the critical question: do certain characteristics reduce the overall risk of death/transmission, or do they merely delay it?

Response: Additional analyses have been done as guided above. Of note there was violation of Cox proportional hazards assumptions by most of the independent variables, thus the accelerated failure time model (ACT) was used as the alternative. Without the use of the Cox proportional hazards model, we could not assess how various covariates influence the hazard rate to help with identifying whether a characteristic reduces the risk or merely delays the event.

Discussion

-- Comparison with Older Data: Comparing the study sample with rural data from 15 years ago may not be the most appropriate choice. Are there more recent studies or datasets, such as World Bank data, that could provide a better comparison? If you decide to compare with the older study, include context about the previous cohort, such as HIV prevalence, maternal nutritional conditions, and other relevant factors.

Response: The two cohorts have a 6-year difference (2009/11 and 2017/19) and not 15 years. Yes, there are recent statistics, but they are more about infant mortality rates and under 5 mortality rates which are not comparable to our 2 years follow up period. We have added context of the 2009/11 cohort in the text as guided.

-- Time in Care and Viral Load: It is surprising that time in care and viral load were not associated with the outcomes. This finding is inconsistent with existing evidence, which shows that longer time in care is associated with reduced neonatal infections and improved survival rates. If your findings diverge from this, it is essential to discuss why this might be the case.

Response: Yes, time in care and viral load were not associated with the main outcome (Child Mortality) since time in care and viral load captured were for the mother and not the child. If these were for the child, then it would be diverging from the existing evidence.

-- Explore better limitations of the study related to selection bias. Another limitation of the study is the potential selection bias introduced by recruiting only mother-infant pairs attending immunization clinics. This approach likely excluded children who were not immunized and who may be at higher risk of mortality from vaccine-preventable diseases. As a result, the study population may represent a healthier subset of the broader community, potentially underestimating the true burden of mortality and transmission in the general population.

Response: Suggestion accepted, and the limitation section of the discussion has been edited to reflect it.

Reviewer #2:

This manuscript describes child mortality rates, up to age 18 months, and associated risk factors, in Uganda, with a particular focus on comparing mortality rates by child HIV exposure status. It provides a valuable contribution to the field of research, particularly as the study period (2017 – 2018) coincided with the Universal ART era - a time marked by increased maternal access to ART and a reduction in vertical HIV transmission rates. Please consider addressing the following comments to help strengthen this manuscript.

Comments to authors:

General

1. General - The comparison of child mortality rates by HIV exposure status appears to be a main focus of this study, however this was not listed as one of the aims. Consider bringing this into the aims.

2. General – Language. Authors should be careful to use non stigmatizing person-first language. Consider infants who are exposed to HIV and infants unexposed to HIV instead of HIV exposed infants and HIV unexposed infants. Consider HIV vertical transmission prevention (VTP) instead of PMTCT.

3. General – Authors should consider carefully reviewing the manuscript for inconsistencies in spaces between words.

Response: Thank you for your time to comprehensively review our manuscript and the constructive feedback you have given. Below are my responses to the comments which have also been reflected in the resubmitted manuscript.

Abstract

4. Abstract – Line 31. Authors should consider adding that they used multivariate regression.

Response: Added.

Introduction

5. Introduction - Line 53: 42 deaths (29-60) per 1,000 live births in 2021. Please specify if the (29-60) is a 95% CI.

Response: Yes, they are the 95% CI. I have deleted them as recommended by reviewer # 1.

6. Introduction – Line 65. Considering including and referencing Evans C, Chasekwa B, Ntozini R, Majo FD, Mutasa K, Tavengwa N, Mutasa B, Mbuya MNN, Smith LE, Stoltzfus RJ, Moulton LH, Humphrey JH, Prendergast AJ; Sanitation Hygiene Infant Nutrition Efficacy (SHINE) Trial Te

---

## [Editor Report · Decision Letter 1]

14 Mar 2025

PONE-D-24-43587R1Mortality rates in a cohort of infants attending immunization clinics in Uganda (2017–2019)PLOS ONE

Dear Dr. Sendagala,

Thank you for submitting your manuscript to PLOS ONE. After careful consideration, we feel that it has merit but does not fully meet PLOS ONE’s publication criteria as it currently stands. Therefore, we invite you to submit a revised version of the manuscript that addresses the points raised during the review process.

We look forward to receiving your revised manuscript.

Kind regards,

Emma K. Kalk

Academic Editor

PLOS ONE

Journal Requirements:

Additional Editor Comments:

“Noted and edited” is not an adequate response to comments. It makes it very time consuming to review the revision. The changes should be included in the table and the new page and line noted.

Please spell out abbreviations the first they appear in both the abstract and manuscript.

Line 35: timing of mother’s ART initiation?

Line 77: are these infants exposed to HIV AND uninfected?

Line 80: does “young female sex” refer to the mothers? In this context I assume that all mothers are female which doesn’t make sense as a risk factor.

Line 96: HEI and HUI haven’t been previously defined.

Line 107: mothers is repeated in brackets.

How was SES determined?

Lines 195-196 and 197-199 are exact repeats of the same sentences.

Line 216: what is the difference between mothers living without a partner and single mothers?

Methods – Line 116. The authors state that they compared the differences in

cumulative survival functions among HEIs and HUIs, however, these results were not

shown in the results section.

Table 3 –The effective sample size was 11514. Can you include this as a footnote?

---

## [Author Response · Author response to Decision Letter 1]

14 Apr 2025

Journal Requirements:

Please review your reference list to ensure that it is complete and correct. If you have cited papers that have been retracted, please include the rationale for doing so in the manuscript text or remove these references and replace them with relevant current references. Any changes to the reference list should be mentioned in the rebuttal letter that accompanies your revised manuscript. If you need to cite a retracted article, indicate the article’s retracted status in the References list and include a citation and full reference for the retraction notice.

Response: All references are up to date. Done changes to

Ref 2: Updated the access link to the current URL (Line 336-337).

Ref 15: Changed to the current updated guidelines of 2021 instead of 2014 (Line 370-372).

Additional Editor Comments:

“Noted and edited” is not an adequate response to comments. It makes it very time consuming to review the revision. The changes should be included in the table and the new page and line noted.

Response: Thank you for guidance. We have edited the responses to match the guidance and added the new line.

Please spell out abbreviations the first they appear in both the abstract and manuscript.

Response: Reviewed the manuscript and all abbreviations spelt out on initial use.

Line 35: timing of mother’s ART initiation?

Response: Deleted “timing of” (Line 36)

Line 77: are these infants exposed to HIV AND uninfected?

Response: They were infants exposed to HIV and uninfected. Added “but not infected” (Line 78)

Line 80: does “young female sex” refer to the mothers? In this context I assume that all mothers are female which doesn’t make sense as a risk factor.

Response: Deleted female sex (Line 82)

Line 96: HEI and HUI haven’t been previously defined.

Response: To match the patient-centered language in the HIV field, deleted HEI and HUI and replaced with “infants exposed and not exposed to HIV” (Line 98)

Line 107: mothers is repeated in brackets.

Response: Deleted the brackets (Line 109)

How was SES determined?

Response: Deleted “socioeconomic status by quintiles” since it was a derived variable (Line 128).

Included a sentence on how the SES was derived (Line 131-132).

Lines 195-196 and 197-199 are exact repeats of the same sentences.

Response: Deleted line 197-199 and retained line 195-196 (currently Line 198-199)

Line 216: what is the difference between mothers living without a partner and single mothers?

Response: There is no difference and for clarity deleted “single mothers” and replaced with “those living with a partner” (Line 222).

Methods – Line 116. The authors state that they compared the differences in

cumulative survival functions among HEIs and HUIs, however, these results were not

shown in the results section.

Response: Results section edited to include the results (Line 210-212)

Table 3 –The effective sample size was 11514. Can you include this as a footnote?

Response: It has been included as a foot note under table 3 (Line 233).

---

## [Editor Report · Decision Letter 2]

22 Apr 2025

Mortality rates in a cohort of infants attending immunization clinics in Uganda (2017–2019)

PONE-D-24-43587R2

Dear Dr. Sendagala,

We’re pleased to inform you that your manuscript has been judged scientifically suitable for publication and will be formally accepted for publication once it meets all outstanding technical requirements.

Kind regards,

Emma K. Kalk

Academic Editor

PLOS ONE
---

## [Editor Report · Acceptance letter]

PONE-D-24-43587R2

PLOS ONE

Dear Dr. Sendagala,

I'm pleased to inform you that your manuscript has been deemed suitable for publication in PLOS ONE. Congratulations! Your manuscript is now being handed over to our production team.

Kind regards,

on behalf of

Dr. Emma K. Kalk

Academic Editor

PLOS ONE